# Control Method of Four Wire Active Power Filter Based on Three-Phase Neutral Point Clamped T-Type Converter

Dawid Buła *, Grzegorz Jarek, Jarosław Michalak and Marcin Zygmanowski

Faculty of Electrical Engineering, Silesian University of Technology, 44-100 Gliwice, Poland; grzegorz.jarek@polsl.pl (G.J.); jaroslaw.michalak@polsl.pl (J.M.); marcin.zygmanowski@polsl.pl (M.Z.)
* Correspondence: dawid.bula@polsl.pl

**Abstract:** An active power filter based on a three-level neutral point clamped T-type converter with LCL input filter is presented in the paper. The main goal of the paper is the analysis of a control system that ensures independent control of a current in each phase. The presented control method of the filter allows reactive power compensation and/or a higher harmonics reduction to be achieved in each phase independently, with the possibility of control tan ($\varphi$) coefficient. This allows the power flow between the phases to be minimalized and reduces the RMS values of filter currents without the need to balance grid currents. The analysis presents the possibility of an operation in different modes, which was verified by experimental results. The results have been obtained in a 20 $A_{RMS}$ laboratory system described in the paper. The results reveal relatively low power losses, which are a feature of the selected three-level T-type topology. Additionally, that topology, when compared to a two-level one, ensures the reduction in current ripples with the same parameters of passive components.

**Keywords:** power quality; three level T-type converter; active power filter; four wire system





## 1. Introduction

Improving the electric power quality is currently the subject of many scientific projects around the world. The problem of electric power quality has become more serious since most of the devices connected to the grid can be described as non-linear loads [1–5]. An example of such a load can be any equipment which uses a diode rectifier with a capacitive filter as a front-end converter. For such a load, the currents flow only when grid voltages are higher than the DC-link capacitor voltage, which causes a high level of current distortions.

Typical devices that consume distorted currents are adjustable speed drives and arc furnaces [5]. At present, the increasing number of single-phase appliances with a diode rectifier input stage is becoming a problem. These are, for example, switch mode power supplies for consumer electronic devices such as modern washing machines, refrigerators, air conditioners, and even energy-saving lighting. In households, public buildings, and companies, there are dozens of nonlinear loads connected to the grid randomly. It can be said that the current distortion sources are, therefore, distributed.

Energy transmission, by means of distorted currents and reactive power, leads to increased losses in the transmission path (switchgear, transformers, and cables). The standards define acceptable levels of total harmonic distortion (THD); therefore, it is necessary to prevent its increase. On the one hand, it stimulates the development of grid-side converters in electrical appliances, which improves the power factor. On the other hand, many non-linear loads are already installed, and it is necessary to counteract their impact on the grid. The available options include passive and active harmonic filters, also referred to as active power filters (APFs) [6–8]. Passive solutions, selected for the specific power and nature of the load, are mainly used with large three-phase loads. Active solutions are more flexible. These are composed of power electronic converters controlled in such a way that the currents are drawn from the grid, being sums of the load and

filter currents, are quasi-sinusoidal and are in phase with the phase voltages, as presented in Figure 1.

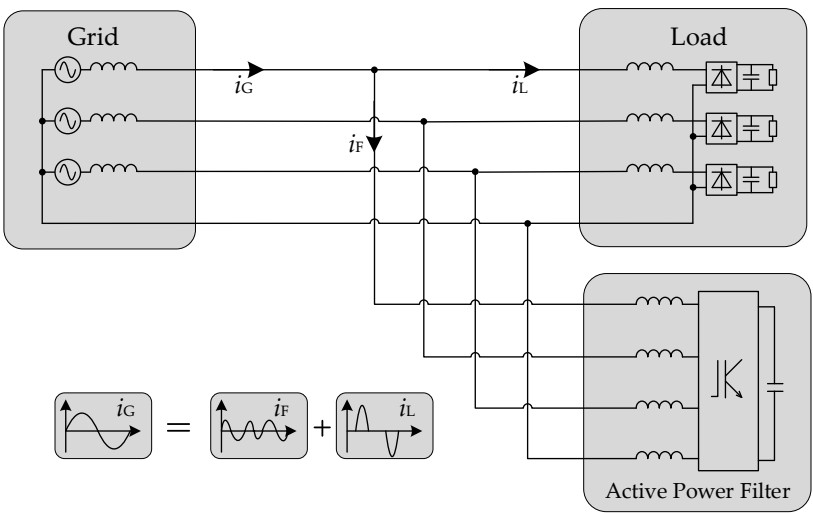

**Figure 1.** Connection of the active power filter to the grid.

The aim of the work, described in the paper, is to develop and test a 20 $A_{rms}$ prototype of shunt active power filter, designed to improve the electrical power quality in office buildings, public utility buildings, or road lighting. The most important requirements for such active power filters are the effectiveness of the operation, low power losses and a small total volume.

Many potential applications require active power filters to work with unbalanced loads. In such case, the currents of each phase must be analyzed independently. With an unbalanced load, the active power filters mainly employ a four-leg converter topology with one leg connected to the neutral wire. Another option is to use a three-leg converter topology with a neutral wire connected to the middle point of the DC-link. One of the most important features of active power filters is their compact design. As a significant part of the volume of active power filters is occupied by passive filter components, particular efforts are made to minimize the required inductance and capacitance values. This can be achieved by increasing the switching frequency and employing multi-level converter topologies.

In this paper, particular attention is paid to the method for determining the reference currents, the control system of the converter, and the structure of the power part of the active power filter.

There are a number of publications on the methods for determining reference currents based on theories of power [9–11], Fourier transform [12–14], direct methods [15–17], artificial neural networks [18–21], and many others [22–25]. In the case of three-phase systems, time-domain methods are most often used. These are the pq [9,25–27] and dq [23,28,29] methods, but their direct application in the proposed solution is not possible because they treat the three-phase circuit as a whole system, and it is not possible to compensate the reactive power independently for each phase without the grid currents balancing. The specificity of these methods means that the sum of the reactive power for the three phases can reach zero value, but in particular phases, the reactive powers will not necessarily be zero. Therefore, it was decided to take a different approach and to treat the three-phase system as three independent single-phase systems. Thus, the dq method for single-phase systems [30–33] was adopted in a three-phase system, with the possibility to choose one of operating modes: reactive power compensation (independently for each phase), reduction in higher harmonics, and simultaneous reactive power compensation and reduction in higher harmonics (power factor improving). A comparison of the various methods for determining the patterns can be found in the work of other authors [3,22,25,26]. The modified dq method was chosen because of the ease of its implementation. It is

characterized by low computational complexity and the possibility of extracting the reactive and higher harmonic components from the load currents, allowing us to achieve the assumed APF operating modes shown in this work.

The second presented aspect is the converter which is employed in the analyzed shunt active power filter. It is based on a three-leg three level NPC type T converter [34,35]. This topology is selected due to its lower power loss generation [36,37] and the ability to reduce current ripples caused by the switching of transistors and common-mode voltage components [38,39]. Both of these features enable us to reduce the size of the converter cooling system and the size of the passive filter, which is a part of the active power filter. The specific feature of the analyzed active power filter is that the neutral wire is connected via a filter inductor to the middle point of the DC-link, as presented in [40]. This solution limits the number of converter legs to three; however, at the cost of the DC-link voltage increase. In ref. [40], the active power filter for four-wire systems based on the three-level NPC topology is presented. In contrast to the classic NPC converter, the NPC type T topology has a lower number of power electronic devices and guarantees lower power losses.

The paper presents a description of the control, i.e., the determination of reference currents and the entire control system, a description of the APF system configuration, and the results of experimental tests confirming the adopted assumptions. Additionally, the paper contains a mathematical proof, confirming the use of a single-phase *dq* transformation to determine the reference currents. These aspects, particularly related to the applied APF topology, are the novel and original contributions.

## 2. Control of Active Power Filter

As it was mentioned, the main feature that was chosen for the designed active power filter is its ability to control currents in each phase independently. Moreover, the APF also allows reactive power only to be controlled, reduces current harmonics, or both compensates reactive power and current harmonics together. This section presents the selected aspects of a control system. Firstly, the idea of a calculation of the reference current in a different mode of operation is described. Then, the description of the whole control system is presented.

### 2.1. Calculation of APF Reference Currents

The fundamental guidelines for the method for determining APF reference currents result directly from APF functionality and from technical and economic restrictions. These are in particular:

- The algorithm should be fast, i.e., the calculations should not take much time due to the delay in the control system—it is necessary to use an algorithm in the time domain.
- The algorithm should allow reactive power compensation and/or higher order harmonics elimination to be achieved independently for each phase, unlike in the case of the most commonly used dq [9,25–27] and pq [23,28,29] methods for three-phase systems, in which the total power is compensated for all phases.
- The algorithm should allow the minimum permissible tangent of phase shift angle $\tan(\varphi)$ to be set.

These guidelines mean that the proposed control method is based on modifications of the dq method for single-phase circuits [30–40]. The starting point of the calculations of the reference currents is the transformation to *dq* synchronously rotating coordinates for each phase independently in the form:

$$\begin{bmatrix} i_d \\ i_q \end{bmatrix} = \begin{bmatrix} \sin(\omega_1 t) & -\cos(\omega_1 t) \\ \cos(\omega_1 t) & \sin(\omega_1 t) \end{bmatrix} \begin{bmatrix} i_\alpha \\ i_\beta \end{bmatrix}, \tag{1}$$

where currents $i_\alpha$ and $i_\beta$ are produced directly from the current for each phase and the current shifted by a quarter of the fundamental period $T$:

$$\begin{bmatrix} i_\alpha \\ i_\beta \end{bmatrix} = \begin{bmatrix} i(t) \\ i(t - T/4) \end{bmatrix}. \tag{2}$$

If the waveform $\cos(\omega_1 t)$ is synchronized with the given phase voltage, both $dq$ current components are composed of the average terms and varying in time terms given as:

$$\begin{bmatrix} i_d \\ i_q \end{bmatrix} = \begin{bmatrix} \bar{i_d} + \tilde{i_d} \\ \bar{i_q} + \tilde{i_q} \end{bmatrix}, \tag{3}$$

It can be stated that:

- The average current component in d-axis is responsible for the fundamental harmonic active power in the given phase. It can be shown that:

$$\bar{i_d} = I_{m(1)} \cos(\varphi_1), \tag{4}$$

- The average component of $i_q$ current is responsible for the fundamental harmonic reactive power in the given phase and can be written as:

$$\bar{i_q} = I_{m(1)} \sin(\varphi_1), \tag{5}$$

- Time varying components of $i_d$ and $i_q$ currents constitute higher harmonics.

By filtering the selected components, one can meet the imposed requirements. In addition, understanding the ratio of constant components, the tangent of the phase-shift angle can be determined:

$$\tan(\varphi_1) = \frac{\bar{i_q}}{\bar{i_d}}, \tag{6}$$

which allows the permissible $\tan(\varphi)_{\max}$ value to be included in the control system (see Equation (7)). Figure 2 shows a diagram of the algorithm for determining the reference current in each phase. Furthermore, in Figure 3, the possible filters of $i_d$ and $i_q$ components, depending on the required functionality of the APF system, have been presented.

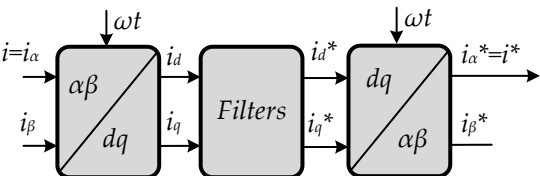

**Figure 2.** A diagram of the algorithm for determining the reference current in each phase.

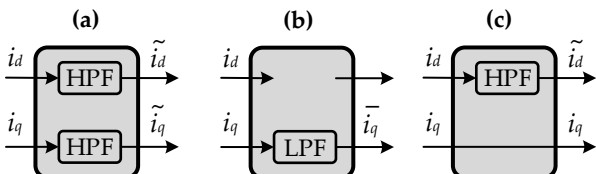

**Figure 3.** Different signal filters for $i_d$ and $i_q$ components, determining the required functionality of the APF system: (**a**) for higher harmonic reduction, (**b**) for reactive power compensation, and (**c**) for both (**a**) and (**b**) purposes.

Possible filters, in the case when the permissible tan $(\varphi)_{max}$ value is included to the control algorithm, have been presented in Figure 4. The reference $i_q^*$, assuming the permissible value of tan $(\varphi)_{max}$ can be expressed by the formula:

$$i_q^* = \bar{i}_q - \min\left\{\bar{i}_d \cdot \text{tg}(\varphi)_{max}, \bar{i}_q\right\}. \tag{7}$$

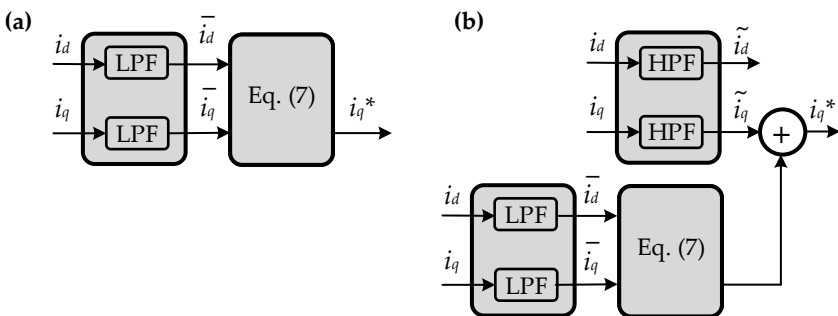

**Figure 4.** Possible filters for $i_d$ and $i_q$ component when the permissible value of tan $(\varphi)$ is included. For the required functionality of the APF system: (**a**) only reactive power compensation and (**b**) higher harmonic elimination with reactive power compensation.

The permissible value of the tangent tan $(\varphi)_{max}$ when multiplied by $\bar{i}_d$ current component defines the level of the permissible average component of the $i_q$ current (reactive power). If the current component $\bar{i}_q$ is lower, the above permissible level of the reference current $i_q^*$ is set to zero. In another case, the permissible level $\bar{i}_d$ tan $(\varphi)$ reduces the value of the reference current $i_q^*$. The presented method is shown for the inductive reactive power but can also be applied to capacitive reactive power with a changed sign of the current.

In the presented method, the signal low-pass filter (LPF) and the high-pass filter (HPF) were used. Their role was to attenuate the average or time-varying components of the *dq* current signals, depending on the operation mode of the APF. These signal filters were mainly responsible for the dynamic response to the load current changes. For ensuring a fast step response, the filter order should not be too high but in other cases, filters had to have an appropriate attenuation rate in the chosen bandwidth. Thus, for the practical realization of the LPF, the second order filter with a cutoff frequency of 16 Hz was chosen. The HPF was realized by subtracting from the input signal and the signal from the LPF filter.

The exact derivation for the dq method for single-phase circuits is presented in Appendix A.

### 2.2. Control System of the APF

Considering reference current calculations presented in Section 2.1, the control system of the APF is presented in Figure 5. The control system consists of three main functions: the detection of the grid voltage space vector angle, the computation of reference currents, and the current control with PMW signal generation.

Based on the measurement of the grid voltages $v_{Ga}$—$v_{Gc}$, the angle of the space vector of the grid voltage was computed by the Double Second Order Generalized Integrator—Frequency Locked Loop (DSOGI-FLL) algorithm [41]. This increased the robustness of the system against the grid voltage harmonics and unbalanced loads. The DSOGI-FLL algorithm computed the amplitude and angle of the first harmonic positive sequence component of the grid voltage $|v_1|$ and $\gamma_a$, respectively. Additionally, for ensuring the transformations of load currents for each converter phase independently, the angle $\gamma_a$ and angles $\gamma_b$, $\gamma_c$ were also computed.

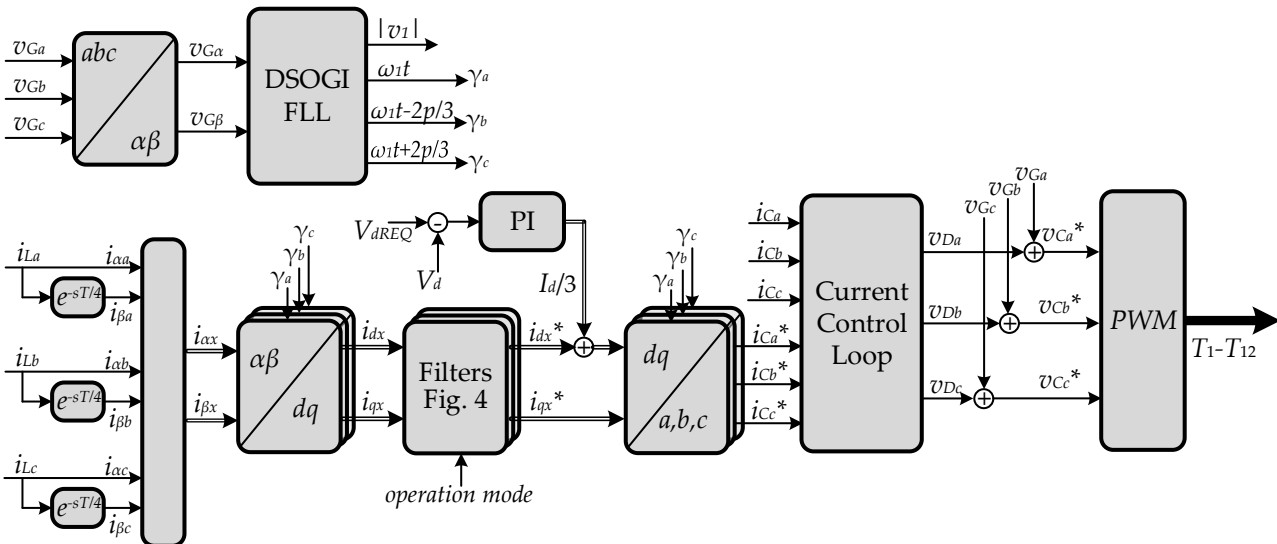

**Figure 5.** Control system of the presented active power filter.

For ensuring independent reduction in the current harmonics and reactive power for each phase, the present value of the phase, load current, and its value delayed by a quarter of the fundamental period (component $e^{-(sT/4)}$) were considered. These signals were treated as phase stationary coordinate systems $i_{\alpha x}$ and $i_{\beta x}$ current components, which formed vectors (where $x = a$, $b$, $c$). A further part of the algorithm of computation of converter reference current consisted of blocks representing three parallel computations for each APF phase independently. Both current components, $i_{\alpha x}$ and $i_{\beta x}$, were transformed to three synchronously rotating coordinate systems using angles $\gamma_a$, $\gamma_b$, and $\gamma_c$. Next, the current components, $i_{dx}$ and $i_{qx}$, were filtered for determining the proper values of reference currents in the synchronously rotating coordinate system. Active power filters can operate in different modes, e.g., harmonics reduction or/and reactive power compensation, and this was presented in the previous section. In a calculation of reference currents, the permissible tangent of the phase shift angle tan ($\varphi$) was taken into account. This is presented in Figure 4 and Equation (7). The output signal from the DC-link PI controller was multiplied by 1/3 and then added to each phase d-axis current components $i_{dx}$. The value 1/3 was taken due to the symmetrical division of the active component into the converter phases. This ensures keeping the DC-link voltage close to required value in steady state and during transients. Next, the current components were transformed back to the stationary coordinates, as presented in Figure 2, and then $\alpha$ components of phase currents $i_{\alpha a}$, $i_{\alpha b}$, and $i_{\alpha c}$, formed reference converter currents $i_{Ca}{}^*$, $i_{Cb}{}^*$, and $i_{Cc}{}^*$.

The reference converter currents $i_{Ca}{}^*$, $i_{Cb}{}^*$, and $i_{Cc}{}^*$ were realized in the current control loop block. This control loop acted similarly to the dead-beat control or model-based predictive control [42] where the filter parameters, converter currents, and switching frequency determined the required value of voltage across the filter. The current control loop operated for each phase ($a$, $b$, and $c$) independently and took into account the measured converter currents $i_{Ca}$, $i_{Cb}$, and $i_{Cc}$ and the inductance of the converter side inductor $L_{1F}$ with adding the grid voltages $v_{Ga}$, $v_{Gb}$, and $v_{Gc}$, forming the reference values of the converter voltages $v_{Ca}{}^*$, $v_{Cb}{}^*$, and $v_{Cc}{}^*$. It had to be noted that a consideration of a full model of the LCL filter would increase the complexity of the control loop. Due to the fact that the converter side inductance $L_{1F}$ was five times higher than the grid side inductor inductance $L_{2F}$ (Table 1), the control loop could be simplified to the operation with a single $L$ type filter. The experimental verification shown in Section 4 revealed that the performed simplification did not significantly influence the operation of the APF. The last block of the control system (Figure 5) was the PWM modulator which generated the gating signals for all transistors of the NPC T-type converter.

**Table 1.** Parameters of an investigated active power filter.

| Parameter | Description | Value |
|:---:|:---:|:---:|
| $V_N$ | Nominal line-to-line voltage RMS | 400 V |
| $I_N$ | Nominal phase current RMS | 20 A |
| $V_{DCN}$ | DC-link nominal voltage | 750 V |
| $f_{sw}$ | Switching frequency | 20 kHz |
| $L_{1F}$ | Converter side filter inductor | 0.75 mH |
| $L_{2F}$ | Grid side filter inductor | 0.15 mH |
| $C_F$ | Filter capacitor | 4.4 µF |
| $C_{DC1}, C_{DC2}$ | DC-link capacitor | 2 mF |

## 3. System Configuration of Active Power Filter

The structure of the proposed APF is depicted in Figure 6. As noted in the introduction, during the selection of the topology of the power circuits, the authors focused on the need to achieve: reduced power losses, the possibility of compensating unbalanced loads, and minimizing the filter volume. To ensure cooperation with unbalanced loads, it was necessary to use a four-leg topology or a three-leg topology with a neutral wire connected to the middle point of the DC-link. Both solutions can be realized with a two-level or three-level converter. It was shown in [36] that the NPC-T converter's feature had a lower level of power losses than in the two-level solution. In the NPC-T topology, as in the two-level converter, there were upper and lower groups of transistors connecting the positive and negative potential of the DC-link with the phase terminals. Additionally, there were transistors in a bidirectional arrangement, connecting the phase terminals with the DC-link midpoint. Dividing the DC-link into two halves provided an easy way to connect the neutral terminal and allowed the filter to be used with unbalanced loads. In the presented application, the integrated version of the circuit in the NPC-T topology was used, which allowed us to reduce the parasitic inductances and correctly select the voltage classes of individual transistors. The split DC-link introduced another advantage in the form of an influence on the value of common mode voltage. The paper [43] presented a significant reduction in the RMS value of the CM voltage generated by the NPC-T topology in relation to the two-level solution.

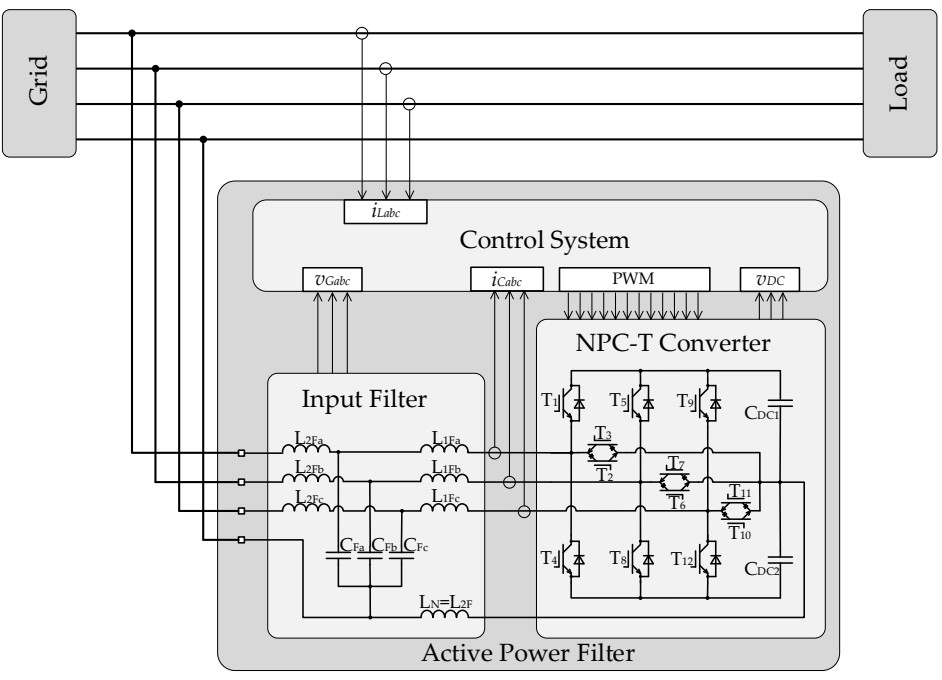

**Figure 6.** Topology of the proposed active power filter.

An LCL filter was used as the input stage before the converter. This type of filter is often used in AC/DC converters due to the greater possibilities of current ripple damping [44]. In the case of a filter used to compensate the distortions introduced by other loads, this property is highly important.

The control algorithm described in Section 2.2 was implemented in the microprocessor controller, based on the Texas Instruments TMS320F28335 microcontroller. It was necessary to measure the currents at two points. The measurement of the converter current (before the filter) and the measurement of the load current had been selected, which must be performed outside of the APF. The grid voltage and shared DC-link voltage measurements were also needed. The chosen converter required preparation of 12 PWM signals. In addition to the implementation of the main algorithm, the controller also provided contactor control, including capacitor precharging and communication, which is not shown in Figure 6.

The parameters of the developed APF prototype, including the values of inductance and capacitance of passive elements, are presented in Table 1. For the realization of the APF prototype, the Fuji Electric module 12MBI50VN-120-50 (1200 V, 50 A) was selected. The switching frequency of 20 kHz was chosen to minimize power losses on one hand, and ensure high dynamics of the filter on the other hand. For the LCL filter, the resonance frequency can be calculated as:

$$f_{res} = \frac{1}{2\pi} \sqrt{\frac{L_{1F}L_{2F}}{(L_{1F} + L_{1F})C_F}} \, , \tag{8}$$

and should be placed in range:

$$\left\langle \frac{k f_N}{0.3} = 6.67 \text{ kHz} < f_{res} < \frac{f_{sw}}{2} = 10 \text{ kHz} \right\rangle, \tag{9}$$

where $k = 40$ is maximum order of harmonics reduced by the APF, $f_N$ = 50 Hz is the fundumental frequency, and $f_{sw}$ is the switching frequency [44].

For the designed LCL filter, the resonance frequency was 6.79 kHz, which met the assumed condition (9). Additionally, the converter side inductor $L_{1F}$ inductance was set to be 5 times higher than the grid side inductor $L_{2F}$ inductance. This limited the converter current ripples to less than 25% of the converter current nominal value, and, as a result, this reduced the inductor $L_{1F}$ power losses.

## 4. Experimental Results

In this section, the operation of the four-wire active power filter is presented. The experimental results are focused on power losses measurements and the functional operation of the APF. The DC-link voltage nominal value of 750 V was needed for the proper operation of APF in a four-wire grid. This was because the neutral wire was connected to the middle point of DC-link capacitor bank. It was necessary that half of the DC-link voltage should be higher than the amplitude of the phase voltage with a margin for proper control of the phase current. The APF in experimental setup during power losses measurement is presented in Figure 7.

### 4.1. Power Losses

The measurement of power losses of the APF was carried out during the reactive power generation mode. Power losses were measured by using the power analyzer WT5000 and were collected in Table 2.

As presented in Figure 8, the power losses $P_{loss}$ range from 72 W, for the minimum value of the line current, up to 447 W, for the rated current of the APF. Higher values of power losses for inductive reactive power generation were caused by using the LCL passive input filter. The capacitor in the LCL filter was responsible for the generation of small inductive reactive power by the APF. This was particularly seen for the generation of zero value reactive power when the power losses of the APF were not equal to zero. Due

to this reason, the power losses for capacitive reactive power generation were smaller than for the inductive reactive power generation. In Figure 9, the characteristic of watt per kvar power losses $R_{loss}$ is shown. One can see that for reactive power higher than 2.5 kvar, the losses $R_{loss}$ are in the range between 28 W/kvar and 45 W/kvar.

**Table 2.** Results of power losses measurement of APF operating in reactive power generation mode.

| $Q_{gen}$, kvar | $I_{LRMS}$, A | $P_{loss}$, W | $R_{loss}$, W/kvar |
|---|---|---|---|
| −13.8 | 20.2 | 441 | 32.0 |
| −11.0 | 16.1 | 327 | 29.7 |
| −8.2 | 12.1 | 236 | 28.8 |
| −5.4 | 7.9 | 162 | 29.9 |
| −2.7 | 4.0 | 102 | 38.0 |
| −0.8 | 1.1 | 74 | 97.7 |
| 0.2 | 0.4 | 72 | 287.3 |
| 0.7 | 1.0 | 76 | 104.2 |
| 2.8 | 4.1 | 113 | 40.6 |
| 5.4 | 8.0 | 173 | 31.9 |
| 8.2 | 12.2 | 257 | 31.2 |
| 10.9 | 16.1 | 345 | 31.6 |
| 13.5 | 20.0 | 447 | 33.2 |

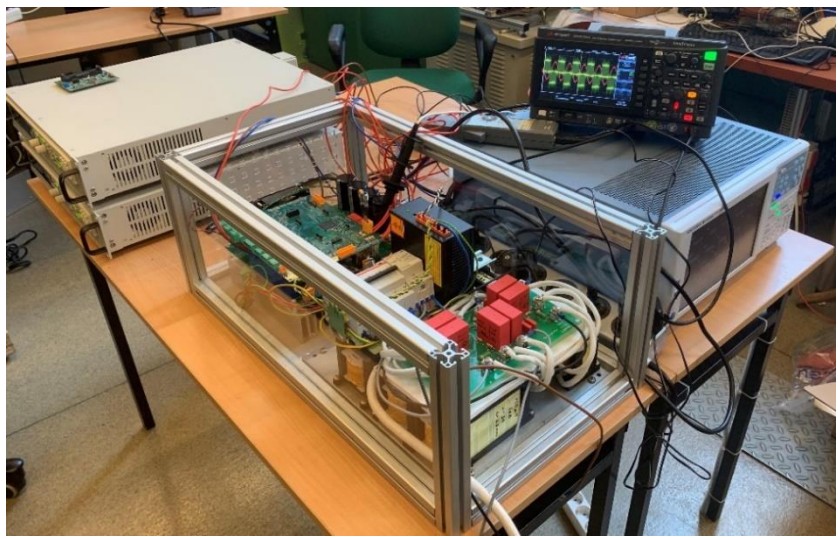

**Figure 7.** Experimental setup for power losses measurement in the three-phase NPC-T converter.

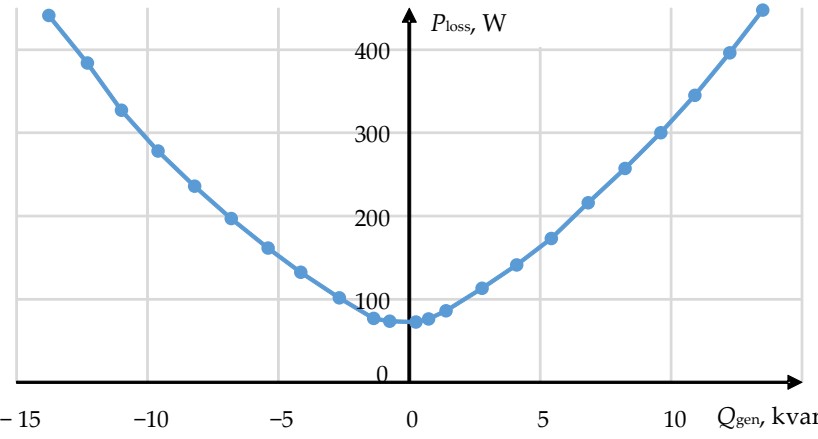

**Figure 8.** Power losses measured during the reactive power generation mode.

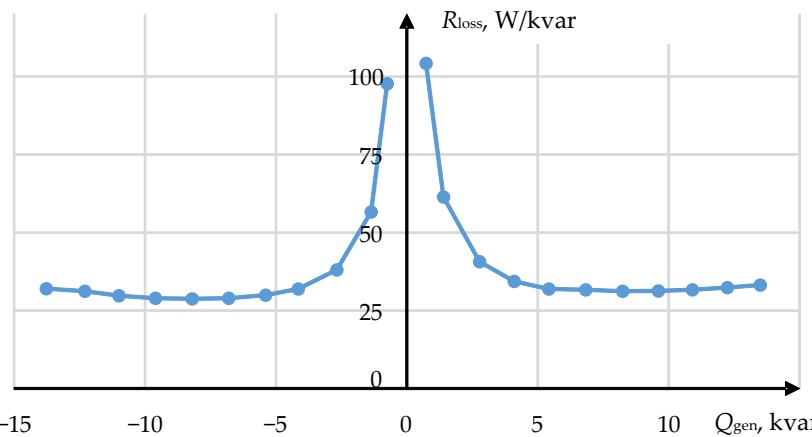

**Figure 9.** Per kvar power losses measured during the reactive power generation mode.

Exemplary results of the operation of the APF during reactive power generation is presented in Figure 10. The results were been obtained using analyzer WT5000. On the left-side, plots of the measured voltages, currents, and powers are presented. On the right-side, plots of the current and voltage waveforms are presented. One can see that the grid voltages are distorted, which caused slight current distortions. Despite these distortions, the phase shift angles between the current and voltage in each phase are satisfactory, and it is also visible that the reactive powers are balanced.

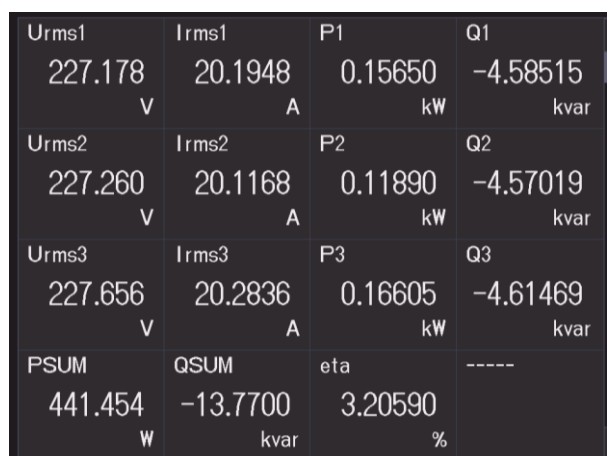
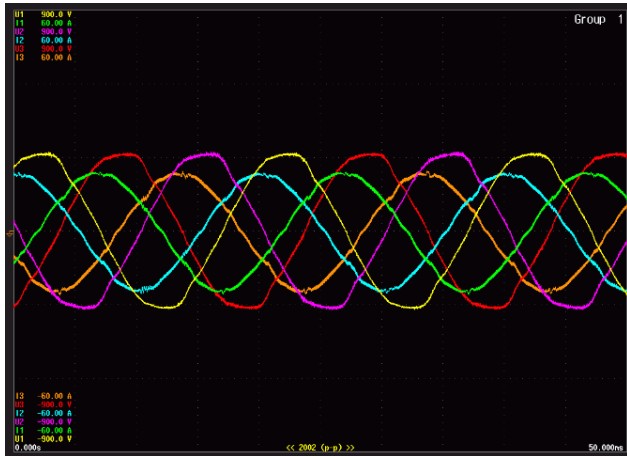

**Figure 10.** Exemplary results for power losses measurement from power analyzer WT5000.

### 4.2. Steady State Operation

As previously mentioned, the APF operated with different currents in each phase without balancing grid currents. The operation with different loads in each phase is presented in Figure 11a. Two nonlinear loads were connected to phase *a* and *b*. The diode rectifier connected to phase *a* was with a resistive-capacitive load at its output. In phase *b*, the diode rectifier, with a series connection of resistor and inductor, was connected. In phase *c*, the series connection of resistor and inductor acted as a linear load example which absorbed the reactive power. The last plot presents the neutral wire current. The harmonic spectrum of the load currents is presented in Figure 11b. It can be seen that currents differ in RMS value and total harmonic distortion coefficient. Effects of active filtering using the presented APF filter are shown in Figure 11c,d. The APF was operating with both the reactive power compensation and harmonics reduction mode. It can be seen that both RMS values of the grid currents and total harmonic distortion coefficient were reduced except

for phase *c*, in which the THD slightly increased due to the operation of the APF. One can see that the APF, as a result of the proposed control system, reduced the harmonics and reactive power without the need for ensuring the grid current balancing, which was proved by different RMS values of the grid currents.

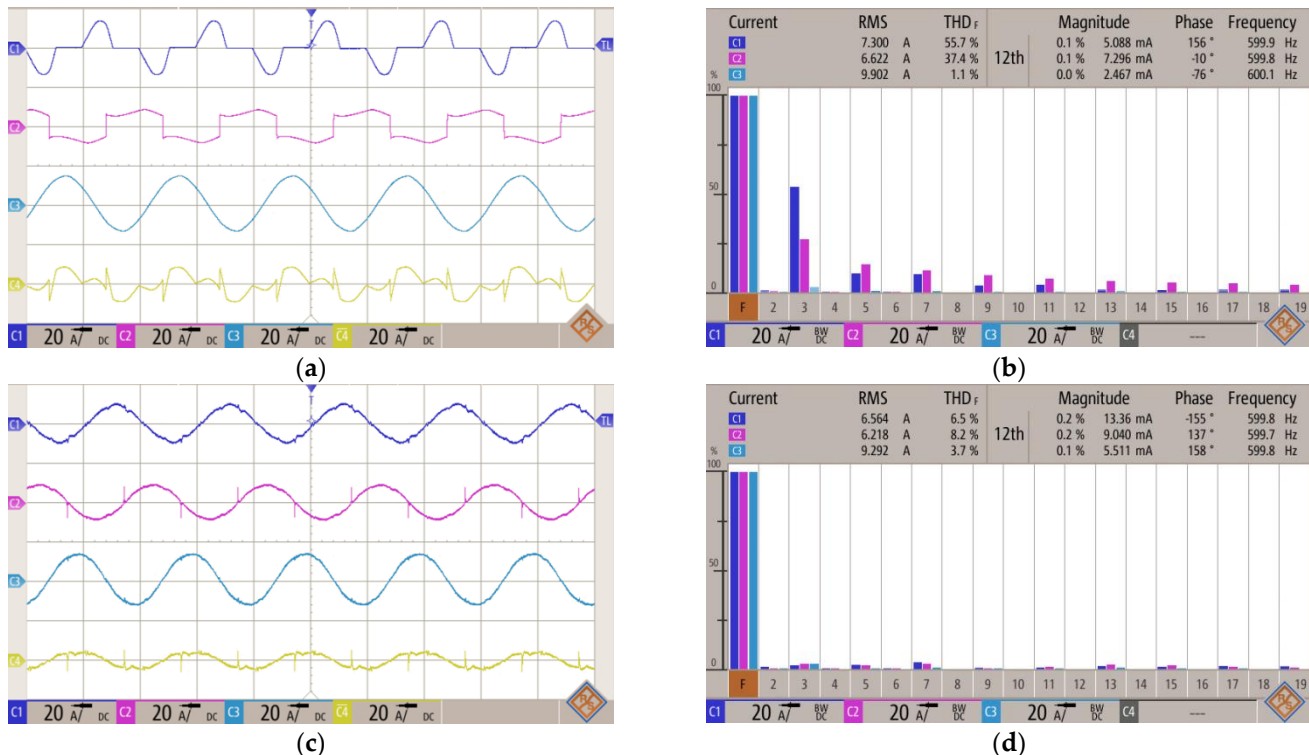

**Figure 11.** Operation of NPC T-type APF with different loads in each phase. Phase *a*—Diode rectifier with RC circuit, phase *b*—Diode rectifier with RL circuit, phase *c*—RL series connection: (**a**) Load side phase and neutral currents; (**b**) Harmonic spectra of load currents; (**c**) Grid side phase and neutral currents; (**d**) Harmonic spectra of grid currents.

The effectiveness of APF, in the case of the operation with typical three-phase balanced nonlinear loads, was analyzed based on the results presented in Figure 12. For testing the operation of the APF, a diode rectifier with parallel RC circuits at its output was treated as a load. The diode rectifier was connected to the grid using the input *L* filter. For ensuring different levels of current THD, different values of the input *L* filter, DC-link capacitor and output resistors were used. Figure 12 presents the load currents (navy blue), the grid currents (pink), and the APF filter currents (blue) in a single phase.

The results obtained for the APF operation with the load current characterized by THD coefficient equal to 26.6% are presented in Figure 12a,b. It can be seen that the grid current harmonics were reduced to THD = 3.3%. The spectrum of harmonics of the load current consisted of typical 5th, 7th, 11th, 13th, etc., harmonics. In the spectrum of the grid current harmonics, the presence of 7th and 13th harmonic was caused by the grid voltage distortions.

The results obtained for the APF operation with the load current characterized by THD coefficient equal to 49.0% and for the load current characterized by THD coefficient 95.2% are presented in Figure 12c–f, respectively. For both situations, the APF filter reduced current harmonics by decreasing THD coefficients of the grid current to 5.3% and 7.9%, respectively. As presented in Figure 12a,c,e, the APF filter currents verified the effectiveness of the APF operation. It can be seen that as a result of the usage of the LCL filter, the switching frequency current components were not so high and the APF guaranteed the high dynamics in current forming, which was verified by a good harmonics reduction.

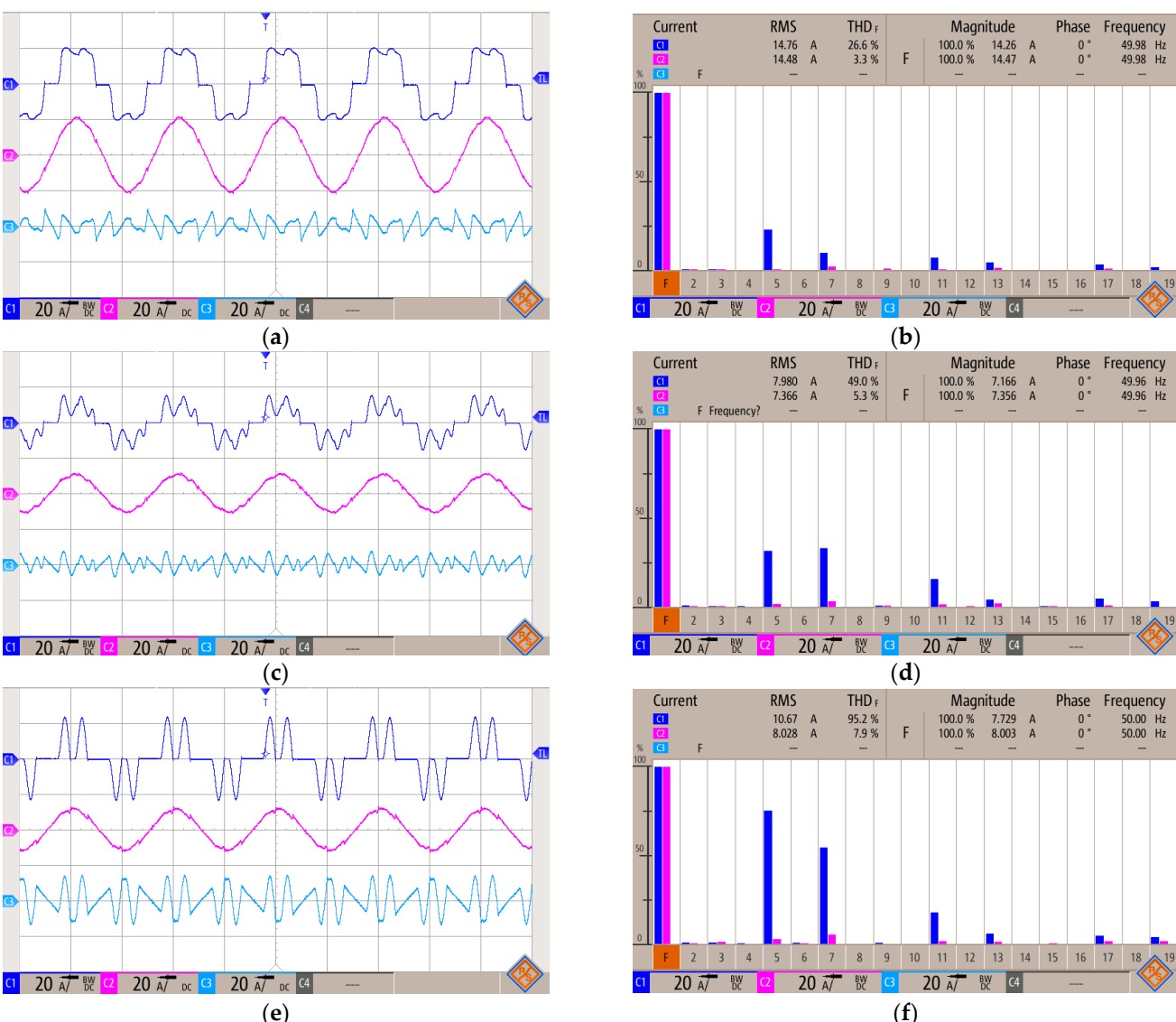

**Figure 12.** Effectiveness of the harmonics reduction for balanced loads—Diode rectifier with RC circuit and input L filters ensuring different THD levels: Load, grid, and APF filter currents for load with 26.6% THD (**a**), 49.0% THD (**c**), and 95.2% THD (**e**), respectively; Harmonic spectra of load and grid currents for load with 26.6% THD (**b**), 49.0% THD (**d**), and 95.2% THD (**f**), respectively.

The different operation modes of the APF are presented in Figure 13. These modes are based on the idea of a calculation of reference current, as presented in Figures 2 and 3. As a load, a three-phase thyristor rectifier with a DC motor was used. Figure 13 presents the load currents (navy blue), the grid voltages (yellow), the grid currents (pink), and the APF filter currents (blue). The operation of the APF for reactive power compensation is presented in Figure 13a. It can be seen that the grid current was in phase with the grid voltage and the APF generated a sinusoidal current. The operation of the APF for harmonics reduction is presented in Figure 13b. It can be seen that the grid current and grid voltage were phase shifted and the APF generated only current harmonics. Due to the limited dynamics of the APF, some peaks in the grid currents can be observed.

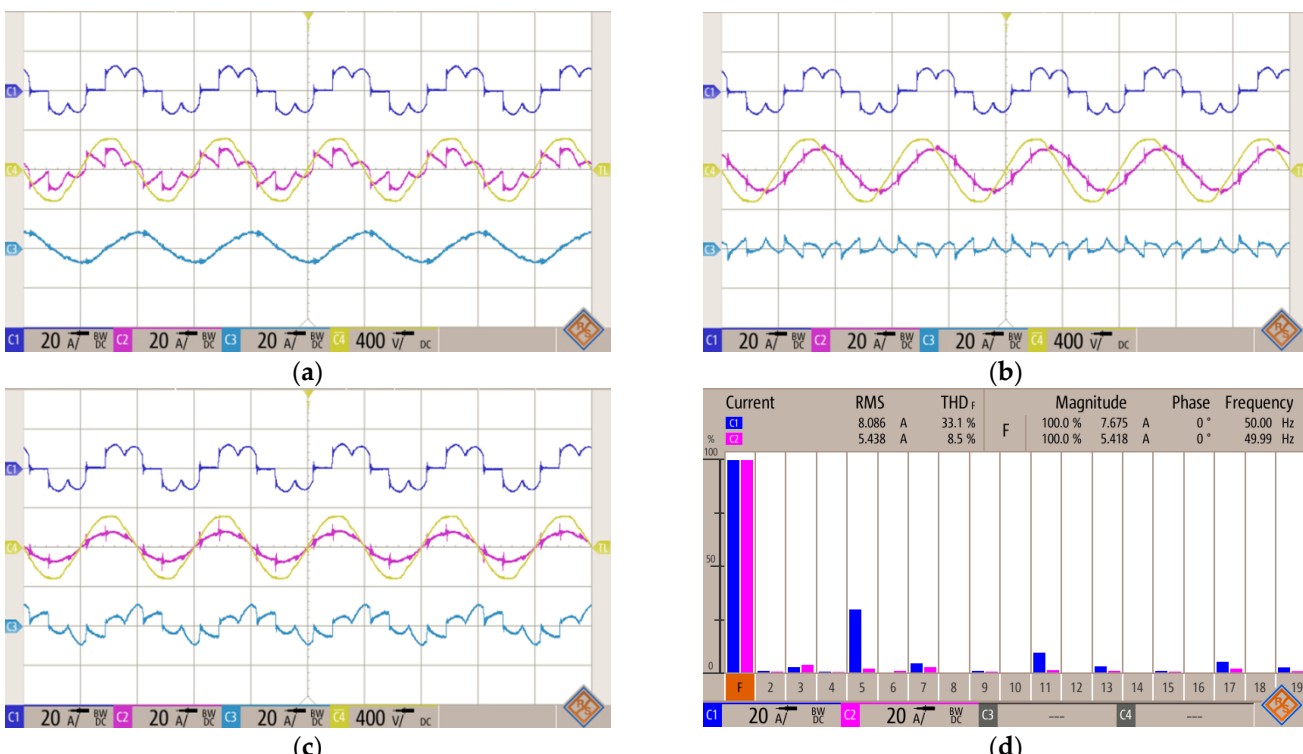

**Figure 13.** Operation of NPC T-type APF in different operation modes (with thyristor rectifier supplying DC motor as a load): (**a**) Operation in reactive power compensation mode; (**b**) Operation in current harmonics reduction mode; (**c**) Operation in simultaneous reactive power compensation and current harmonics mode; (**d**) Harmonic spectra of load and grid currents for simultaneous operation mode.

The operation of the APF for simultaneous harmonics reduction and reactive power compensation is presented in Figure 13c. It can be seen that the grid current is quasi-sinusoidal with no phase shift between the grid voltage and current with small peaks caused by the limited dynamics of the APF. It can be seen that for this operation mode, the RMS value of the APF current is maximal. The spectrum of the load current and grid current harmonics for the third mode of operation is presented in Figure 13d. It is seen that the grid current is composed of 3rd (caused by the control system of APF), 7th, and 17th harmonics (caused by the grid voltage distortions). It should be mentioned that the APF filter control system ensured proper operation with loads, characterized by both power dissipation and recuperation in each mode and ensured the realization of required tan ($\varphi$).

### 4.3. Dynamic Response of the APF

The dynamic reaction of the APF on switching on the load is presented in Figure 14. After connecting the load to the grid, the APF did not react for a time close to 10 ms. It was caused by the idea of an independent control of the reference current in each phase, which needed a 5 ms time delay to compute and filter dq current components in each phase. It can be seen that the proper reaction of the APF took about 30 ms which was connected using a control system low pass filter with a cutoff frequency of 16 Hz. After that time, the control system properly computed the reference currents and ensured the proper operation of the active power filter.

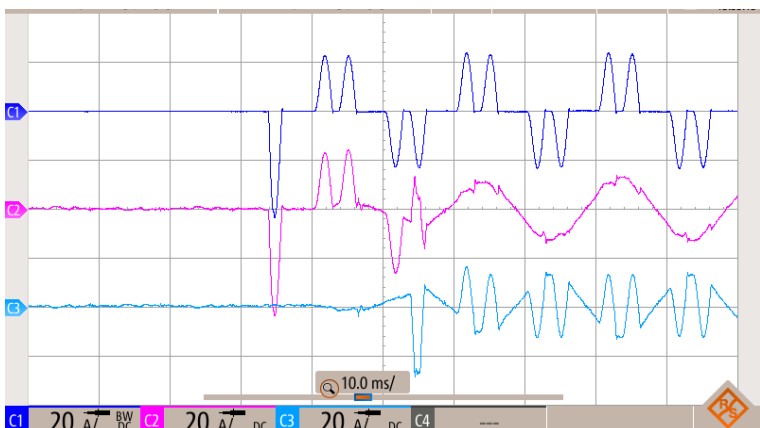

**Figure 14.** Operation of NPC T-type APF during transient.

## 5. Conclusions

The achieved results proved the proper operation of the presented three-level neutral point clamped T-type converter based on an active power filter. The proposed control system of the APF allowed us to compensate for the reactive power and/or reduce higher current harmonics. The control system ensured the realization of such functions independently in each phase, without the need to balance the grid currents; this was experimentally verified. The proposed control system was based on the dq method used in single-phase systems and was widely presented in Section 2 and Appendix A. Moreover, the control method was able to reduce the generated current by taking into account the permissible tan ($\varphi$). The dynamics of the applied control system was mainly limited by the cutoff frequency of used LPF filters and proposed phase reference current calculations and results in 30 ms reaction for step change of the load.

As presented, the three-level neutral point clamped T-type converter ensured relatively low power losses and reduction in current ripple in comparison to the two-level topology. The designed LCL filter ensured satisfying current ripple damping with active power filter high dynamics. The used division of inductance in the LCL filter ensured the limitation of current ripples in the converter side inductor and ensured the possibility of the current control loop's simplification without the need to use additional signal measurements and control loops.

**Author Contributions:** Conceptualization, D.B. and J.M.; methodology, D.B., G.J., J.M. and M.Z.; software, D.B. and J.M.; validation, D.B., G.J., J.M. and M.Z.; formal analysis, D.B. and J.M.; writing—original draft preparation, D.B., G.J., J.M. and M.Z.; writing—review and editing, D.B., G.J., J.M. and M.Z.; supervision, D.B. and J.M. All authors have read and agreed to the published version of the manuscript.

**Funding:** This research received no external funding.

**Institutional Review Board Statement:** Not applicable.

**Informed Consent Statement:** Not applicable.

**Data Availability Statement:** Data sharing not applicable.

**Conflicts of Interest:** The authors declare no conflict of interest.

## Appendix A

Single-phase dq transformation.
Assuming that the phase current can be written as a Fourier series:

$$i(t) = \text{Re} \sum_{h \in N} I_{(h)} e^{jh\omega_1 t}, \tag{A1}$$

where $I_{(h)} = I_{m(h)}e^{j\varphi_{(h)}}$—amplitude complex value for a specific harmonic—then we can transform it into a form:

$$i(t) = \frac{1}{2}\left(\sum_{h \in N} I_{(h)}e^{jh\omega_1 t} + \sum_{h \in N} I^*_{(h)}e^{-jh\omega_1 t}\right). \tag{A2}$$

Then, assuming the current in the $\alpha\beta$ system:

$$I_{\alpha\beta}(t) = i_\alpha + ji_\beta, \tag{A3}$$

and in this case:

$$i_\alpha = i(t), \tag{A4}$$

$$i_\beta = i(t - T/4), \tag{A5}$$

So, the $\alpha\beta$ components of this current are equal to:

$$i_\alpha = \frac{1}{2}\left(\sum_{h \in N} I_{(h)}e^{jh\omega_1 t} + \sum_{h \in N} I^*_{(h)}e^{-jh\omega_1 t}\right), \tag{A6}$$

$$i_\beta = \frac{1}{2}\left(\sum_{h \in N} I_{(h)}e^{jh\omega_1 t}e^{-jh\frac{\pi}{2}} + \sum_{h \in N} I^*_{(h)}e^{-jh\omega_1 t}e^{jh\frac{\pi}{2}}\right), \tag{A7}$$

which can be expressed as:

$$\begin{aligned}I_{\alpha\beta}(t) = {} & \frac{1}{2}\left(\sum_{h \in N} I_{(h)}e^{jh\omega_1 t} + \sum_{h \in N} I^*_{(h)}e^{-jh\omega_1 t}\right) \\ & + j\frac{1}{2}\left(\sum_{h \in N} I_{(h)}e^{jh(\omega_1 t - \frac{\pi}{2})} + \sum_{h \in N} I^*_{(h)}e^{-jh(\omega_1 t - \frac{\pi}{2})}\right).\end{aligned} \tag{A8}$$

Applying the *dq* transformation in the form:

$$I_{dq}(t) = i_d + ji_q = I_{\alpha\beta}(t)e^{-j\omega_1 t}, \tag{A9}$$

we get:

$$\begin{aligned}I_{dq}(t) = {} & \frac{1}{2}\left(\sum_{h \in N} I_{(h)}e^{jh\omega_1 t}e^{-j\omega_1 t} + \sum_{h \in N} I^*_{(h)}e^{-jh\omega_1 t}e^{-j\omega_1 t}\right) \\ & + j\frac{1}{2}\left(\sum_{h \in N} I_{(h)}e^{jh(\omega_1 t - \frac{\pi}{2})}e^{-j\omega_1 t} + \sum_{h \in N} I^*_{(h)}e^{-jh(\omega_1 t - \frac{\pi}{2})}e^{-j\omega_1 t}\right),\end{aligned} \tag{A10}$$

This can be expressed as:

$$\begin{aligned}I_{dq}(t) = {} & I_{(1)} + \frac{1}{2}\left(\sum_{h > 1} I_{(h)}e^{jh\omega_1 t}e^{-j\omega_1 t} + \sum_{h \in N} I^*_{(h)}e^{-jh\omega_1 t}e^{-j\omega_1 t}\right) \\ & + j\frac{1}{2}\left(\sum_{h > 1} I_{(h)}e^{jh(\omega_1 t - \frac{\pi}{2})}e^{-j\omega_1 t} + \sum_{h \in N} I^*_{(h)}e^{-jh(\omega_1 t - \frac{\pi}{2})}e^{-j\omega_1 t}\right).\end{aligned} \tag{A11}$$

By dividing into average and time-varying components:

$$I_{dq}(t) = \overline{I_{dq}} + \widetilde{I_{dq}}, \tag{A12}$$

we can write:

$$\overline{I_{dq}} = I_{(1)}, \tag{A13}$$

$$\widetilde{I_{dq}} = \frac{1}{2}\left(\sum_{h>1} I_{(h)}e^{jh\omega_1 t}e^{-j\omega_1 t} + \sum_{h\in N} I_{(h)}^* e^{-jh\omega_1 t}e^{-j\omega_1 t}\right)$$

$$+j\frac{1}{2}\left(\sum_{h>1} I_{(h)}e^{jh(\omega_1 t-\frac{\pi}{2})}e^{-j\omega_1 t} + \sum_{h\in N} I_{(h)}^* e^{-jh(\omega_1 t-\frac{\pi}{2})}e^{-j\omega_1 t}\right), \tag{A14}$$

So, the average component (A13) is directly related to the active and reactive components of the current of the first harmonic:

$$\overline{I_{dq}} = \overline{i_d} + j\overline{i_q} = I_{m(1)}\cos(\varphi_1) + jI_{m(1)}\sin(\varphi_1). \tag{A15}$$

By filtering out the average and time-varying *dq* components (Figure 3a), we can obtain specific components. For example, by filtering out the average component, we will get after the inverse transform:

$$I_{\alpha\beta}(t) = \widetilde{I_{dq}}e^{j\omega_1 t}, \tag{A16}$$

which gives:

$$I_{\alpha\beta}(t) = \frac{1}{2}\left(\sum_{h>1} I_{(h)}e^{jh\omega_1 t} + I_{(1)}^* e^{-j\omega_1 t} + \sum_{h>1} I_{(h)}^* e^{-jh\omega_1 t}\right)$$

$$+j\frac{1}{2}\left(\sum_{h>1} I_{(h)}e^{jh(\omega_1 t-\frac{\pi}{2})} + I_{(1)}^* e^{-j(\omega_1 t-\frac{\pi}{2})}\right.$$

$$\left. + \sum_{h>1} I_{(h)}^* e^{-jh(\omega_1 t-\frac{\pi}{2})}\right), \tag{A17}$$

where:

$$I_{(1)}^* e^{-j\omega_1 t} + jI_{(1)}^* e^{-j(\omega_1 t-\frac{\pi}{2})} = I_{(1)}^* e^{-j\omega_1 t} + I_{(1)}^* e^{-j\omega_1 t}e^{j\pi} = 0, \tag{A18}$$

so:

$$I_{\alpha\beta}(t) = \frac{1}{2}\left(\sum_{h>1} I_{(h)}e^{jh\omega_1 t} + \sum_{h>1} I_{(h)}^* e^{-jh\omega_1 t}\right)$$

$$+j\frac{1}{2}\left(\sum_{h>1} I_{(h)}e^{jh(\omega_1 t-\frac{\pi}{2})} + \sum_{h>1} I_{(h)}^* e^{-jh(\omega_1 t-\frac{\pi}{2})}\right). \tag{A19}$$

and finally:

$$I_{\alpha\beta}(t) = i_\alpha + ji_\beta = \operatorname{Re}\sum_{h>1} I_{(h)}e^{jh\omega_1 t} + j\operatorname{Re}\sum_{h>1} I_{(h)}e^{jh(\omega_1 t-\frac{\pi}{2})}. \tag{A20}$$

So there are only higher harmonic components in the reference current ($i^* = i_\alpha$). By filtering out only the average component in the current $i_d$ (Figure 3c), we will get:

$$I_{\alpha\beta}(t) = i_\alpha + ji_\beta = I_{m(1)}\sin(\varphi_1)e^{j\omega_1 t} + \operatorname{Re}\sum_{h>1} I_{m(h)}e^{jh\omega_1 t}$$

$$+j\operatorname{Re}\sum_{h>1} I_{m(h)}e^{jh(\omega_1 t-\frac{\pi}{2})}, \tag{A21}$$

So:

$$i_\alpha = I_{m(1)}\sin(\varphi_1)\cos(\omega_1 t) + \operatorname{Re}\sum_{h>1} I_{m(h)}e^{jh\omega_1 t}. \tag{A22}$$

Assuming that the signal $\cos(\omega_1 t)$ is synchronous with the voltage, the $i_\alpha$ current has a component related to reactive power and higher harmonics ($i^* = i_\alpha$).

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
