# Peer review of "Control Method of Four Wire Active Power Filter Based on Three-Phase Neutral Point Clamped T-Type Converter"

_energies, doi:10.3390/en14248427_

Round 1

Reviewer 1 Report

According to the authors' declaration, which is presented in Abstract: line about 12, and in the Introduction: line about 53 and 83-94, the paper focuses on the partial compensation of the inactive load current - in this case without the balancing of the power line phase currents, even if the load is unbalanced (lines 223-240).

Such a method of nonactive current compensation may be considered controversial, but it undoubtedly reduces the power losses of the active filter during compensation. From this point of view, such work of the active filter can be regarded as economically justified from the point of view of the filter user.

In the opinion of this reviewer, the authors clearly and specifically guide the reader through the issues raised. Only minor remarks (and impressions) have been formulated below, which – if considered – in the opinion of this reviewer may contribute to better understanding of the paper.

Line about 56. The phase currents do not must, although they may, be analyzed separately. There are solutions in which the phase currents of the load are only tracked in order to obtain the inactive current reference that is understood as the difference of the total current to its active component.

This pattern is then generated by the active filter as a compensating current, often requiring the power line phase currents to be balanced.

Lines 72-80. It is obvious to all of us that authors who consider a three-phase circuit to be "energetically uniform" are right and they would have no difficulty in designing an active filter according to the ideas presented in this article. It is their deliberate choice, based on the assumption not to "transform" the energy losses resulting from the load asymmetry on the supply line side.

Line 248.  “chapter 2.2” -> Section 2.2 (?)

Line 283 and 284.  “Power loses”  -> Power losses (?)

Line about 287-290.  “power loss” -> power losses  (?, it’s about consistency in nomenclature used)

Line 316. From the perspective of the declared operation of the filter, wouldn't it be better to replace “…can operate … without a need for balancing the grid currents” with the phrase “… can operate … without balancing grid currents”?

Line 323.  “… currents differs …”  ->  “… currents differ …” (?)

Lines 331-332 and Figure 12.  From what point of view are the phase currents of the load symmetrical?

Section 4.2.   Active filter operation (source current bettering) for permanent transients?

In opinion of this reviewer the paper is good. The Authors of the paper are free to take into account the opinions of this reviewer. They are merely a form of pre-publication discussion of issues of an area of our common interest.

Author Response

The authors would like to thank the Reviewer for the effort in reviewing the paper.

According to the authors' declaration, which is presented in Abstract: line about 12, and in the Introduction: line about 53 and 83-94, the paper focuses on the partial compensation of the inactive load current - in this case without the balancing of the power line phase currents, even if the load is unbalanced (lines 223-240).

Such a method of nonactive current compensation may be considered controversial, but it undoubtedly reduces the power losses of the active filter during compensation. From this point of view, such work of the active filter can be regarded as economically justified from the point of view of the filter user.

Response: Thank you for your opinion. As you mention, taking into account both partial compensation of inactive currents and operation without line phase currents balancing was dictated by economical aspects.

In the opinion of this reviewer, the authors clearly and specifically guide the reader through the issues raised. Only minor remarks (and impressions) have been formulated below, which – if considered – in the opinion of this reviewer may contribute to better understanding of the paper.

Line about 56. The phase currents do not must, although they may, be analyzed separately. There are solutions in which the phase currents of the load are only tracked in order to obtain the inactive current reference that is understood as the difference of the total current to its active component.

This pattern is then generated by the active filter as a compensating current, often requiring the power line phase currents to be balanced.

Response: Thank you for your suggestion. It has been changed in the final version of the paper

Lines 72-80. It is obvious to all of us that authors who consider a three-phase circuit to be "energetically uniform" are right and they would have no difficulty in designing an active filter according to the ideas presented in this article. It is their deliberate choice, based on the assumption not to "transform" the energy losses resulting from the load asymmetry on the supply line side.

Response: We can only agree with the above statement

Line 248.  “chapter 2.2” -> Section 2.2 (?)

Line 283 and 284.  “Power loses”  -> Power losses (?)  

Line about 287-290.  “power loss” -> power losses  (?, it’s about consistency in nomenclature used).

Line 316. From the perspective of the declared operation of the filter, wouldn't it be better to replace “…can operate … without a need for balancing the grid currents” with the phrase “… can operate … without balancing grid currents”?

Line 323.  “… currents differs …”  ->  “… currents differ …” (?)

Lines 331-332 and Figure 12.  From what point of view are the phase currents of the load symmetrical?

Section 4.2.   Active filter operation (source current bettering) for permanent transients?

Response: The suggested amendments have been included in the final version of the paper. In addition, the paper has undergone a linguistic correction.

In opinion of this reviewer the paper is good. The Authors of the paper are free to take into account the opinions of this reviewer. They are merely a form of pre-publication discussion of issues of an area of our common interest

Reviewer 2 Report

Dear Editor and Authors, I read with interest the manuscript and I will start by saying that is well written.  

The manuscript provides experimental results to support the theory. I have the following questions regarding the applicability of your method. 1) How is your approach implemented to topologies without available neutral (e.g. three-phase H-bridge)? 2) How do you balance the CDC1 and CDC2 capacitors

Author Response

The authors would like to thank the Reviewer for the effort in reviewing the paper.

Dear Editor and Authors, I read with interest the manuscript and I will start by saying that is well written.  

The manuscript provides experimental results to support the theory. I have the following questions regarding the applicability of your method.

1) How is your approach implemented to topologies without available neutral (e.g. three-phase H-bridge)?

Response: Our approach can be implemented into three-wire system fully for reducing harmonics only. Reactive power compensation with the proposed control method requires the presence of the neutral wire. When the converter utilizes the 2-level topology (also referred as H-bridge) our control method requires splitting the dc-link into two capacitors with the middle point connected to the neutral wire. For such a converter it is required to use PWM technique without any zero-sequence components like space vector modulation SVM or PWM with third harmonic injection.   

2) How do you balance the CDC1 and CDC2 capacitors

Response: Dc link capacitor voltage balancing in the proposed APF was out of the scope of the paper. We plan to publish another paper showing the voltage balance method in more detail. Here we mention only the general idea of the applied balancing method.

It should be noted that in the presented NPC-T converter only a dc current component flowing in the neutral wire influences dc-link middle point unbalance. Also, other events can cause voltage unbalance among which there are transients during switching the loads. For preventing voltage unbalance the following steps have been taken: 1) in measurement circuits current transformers and high pass filters are used, 2) both dc-link voltages are measured, and their measured values are applied into PWM unit separately, 3) two high-resistance resistors parallelly connected to dc-link capacitors are used, 4) a low dynamic PI controller reducing dc-link voltage unbalance by a zero-sequence current injection can be also applied.

Reviewer 3 Report

The paper presents a control methodology for active filters to reactive power compensation and power quality improvement. The Paper is well written and easy to understand. However, I would recommend authors to mention why their dq model-based control algorithm is better than other control methodologies described very briefly in the introduction section. The performance of the filter using the proposed 3 leg topology and control algorithm is not compared with already published work. Also please clearly indicate the main novelty of the paper, is it the algorithm or the 3 leg topology?

Author Response

The authors would like to thank the Reviewer for the effort in reviewing the paper.

The paper presents a control methodology for active filters to reactive power compensation and power quality improvement. The Paper is well written and easy to understand.

However, I would recommend authors to mention why their dq model-based control algorithm is better than other control methodologies described very briefly in the introduction section.

Response: The aim of the article was not to compare control methods. Such a comparison can be found in the works of other authors [3, 22, 25, 26] and it would be unreasonable to duplicate it. The modified dq method has been chosen because of the ease of its implementation. It is characterized by low computational complexity and the possibility of isolating the reactive and higher harmonic components from load currents. This allowed achieving the assumed APF operating modes. The above has been added to the introduction.

The performance of the filter using the proposed 3 leg topology and control algorithm is not compared with already published work.

Response: In authors opinion it is very hard to make such comparison. This can be done meaningfully by testing different configurations under the same conditions. However, it is not possible to relate the obtained results directly to the results of other authors. For example, we cannot say that in other works the THD of current was achieved at the level of 4% and our system reduces THD to 3.3% and therefore it is better. The presented solution shows, above all, a specific control algorithm that allows the system to work in various modes (THD reduction, reactive power compensation). The T-type NPC topology selected for implementation for a 4-wire network ensures low power losses and small current ripples at relatively low cost.

Also please clearly indicate the main novelty of the paper, is it the algorithm or the 3 leg topology?

Response: In authors’ opinion both the topology and the algorithm are novelty of the paper. The proposed approach allows for operation in various modes with the assumed tg(φ). The authors have not found similar solutions among numerous publications. Also, the configuration of the APF in the
T-type NPC topology for a 4-wire network was not found.
In response to the review, the content of the article was supplemented with the emphasis on novelty of the solution in accordance with the above.

Round 2

Reviewer 2 Report

Thank you for adding new content. The Authors have addressed the Reviewer's comments.